# Expectation Values of the Neutral Chromium Radius

**Nafeesah Abdul Rahim Yaqub [1], Rabia Qindeel [1]** 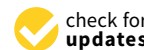**, Norah Alonizan [2]** **and Nabil Ben Nessib [1,3,\*]**

[1] Department of Physics and Astronomy, College of Sciences, King Saud University, PO Box 2455, Riyadh 11451, Saudi Arabia; 437203488@student.ksu.edu.sa (N.A.R.Y.); rqindeel@ksu.edu.sa (R.Q.)

[2] Department of Physics, College of Science, Imam Abdulrahman Bin Faisal University, P.O. Box 1982, Dammam 31441, Saudi Arabia; nalonizan@iau.edu.sa

[3] Groupe de Recherche en Physique Atomique et Astrophysique (GRePAA), INSAT, Centre Urbain Nord, BP 676, University of Carthage, 1080 Tunis Cedex, Tunisia

\* Correspondence: nbennessib@ksu.edu.sa; Tel.: +966-114699737

**Abstract:** Neutral Chromium (Cr I) is an important element in many laboratory plasma applications. In this work, expectation values of the radius for Cr I are calculated. These atomic data are calculated with three different atomic codes: Cowan code using the Hartree–Fock Relativistic approximation, SUPERSTRUCTURE and AUTOSTRUCTURE codes using scaled Thomas–Fermi–Dirac–Amaldi potential. Relativistic corrections are introduced according to the Breit–Pauli approach. The $3d^5 4s$, $3d^4 4s^2$, $3d^5 4d$, $3d^5 4p$ and $3d^4 4s 4p$ configurations are included to obtain the expectation values of radius of Cr I and compared with available data. The novelty of our work is to obtain new values of $< \frac{1}{r} >, < r >$, and $< r^2 >$ for the configuration of $4p$ and $4d$ and the values of $< r^3 >$ for all orbitals configurations considered in this work.

**Keywords:** neutral chromium; radius expectation; powers of the orbitals radius; Hartree–Fock Relativistic approximation; Thomas–Fermi–Dirac–Amaldi potential

## 1. Introduction

The Chromium (Cr I) is a transition metal and has electronic configuration as [Ar] $3d^5 4s^1$, ground-state level: $^7S^3$ and ionization energy: 6.76651 eV (National Institute of Standards and Technology (NIST) database [1]). Description and analysis of the first spectrum of neutral chromium (Cr I) have been studied experimentally by Kiss 1953 [2]. Wavelengths and estimated intensities are presented for about 4400 lines of Cr I recorded photographically between 11,610 Å in the infrared and 1880 Å in the ultraviolet.

For laboratories and astrophysical plasmas, the study of chromium is important and it occurs in some fusion experiments. Many laboratory techniques are used to have Cr I atomic structure data (see Sobeck et al. 2007 [3]). Chromium in its various ionization stages is important for analyzing atmospheres of some stars and many works are done for the chromium in astrophysical plasmas (see [4]).

The powers of the orbital radius of elements are needed in many cases. As example, for Stark broadening calculations, to obtain widths and shifts of a given line, we need as input the energy levels and oscillator strengths, $< r >$ and $< r^2 >$ of the initial, final and all the perturbing levels (see [5–7]). The contribution of the dipolar interaction in the Stark broadening is directly proportional to square orbital radius and a variation of it can affect directly the result of widths and shifts of the spectral lines broadened by Stark effect. Orbital radius and their square are also in the contribution of the quadrupolar interaction.

A sound knowledge of Cr I physical properties is essential, thus the main objective of this study is to calculate expectation values of Cr I radius.

## 2. Calculation Methods

Radius expectation values for the different orbitals of neutral chromium atom are carried out with two different methods:

In the first method, the Hartree–Fock Relativistic (HFR) approximation have been applied using the Cowan (CW) atomic structure code [8] by intermediate coupling schemes and the following $3d^54s, 3d^44s^2, 3d^54d$ even and $3d^54p, 3d^44s4p$ odd configurations containing six active electrons. The radial integrals $F^k$ were kept at 85%, the exchange integral $G^k$ and configuration interaction integrals $R^k$ were kept at 80% of the HFR values. These ratios are normally used for reasonably good ab initio predictions [8].

In the second method, the scaled Thomas–Fermi–Dirac–Amaldi (TFDA) potential has been applied using SUPERSTRUCTURE (SS) and AUTOSTRUCTURE (AS) atomic structure codes. Relativistic corrections are introduced according to the Breit–Pauli approach, SS and AS codes are used with the same first five excitation configurations. The SS code was originally developed by Eissner et al. 1974 [9] and later modified by Nussbaumer and Storey 1978 [10] to ensure greater flexibility in the radial orbital functions. The AS code is an extension of the SS code incorporating different relativistic corrections of the Hamiltonian(Badnell, 1997 [11]).

The radial functions are calculated in scaled Thomas–Fermi statistical model potential which depends on parameters called $\lambda_{nl}$. Those $n$ and $l$ dependent scaling parameters $\lambda_{nl}$ were determined variationally by minimizing the sum of the energies of all the target terms. The scaling parameters $\lambda_{nl}$ obtained in the SS and AS atomic structure codes are given in Table 1. The difference between the values is caused by a small change in the optimization procedure in AS compared to the SS procedure.

**Table 1.** Scaling parameters $\lambda_{nl}$ used in SUPERSTRUCTURE and AUTOSTRUCTURE atomic structure codes.

| Orbital | 1s | 2s | 2p | 3s | 3p | 3d | 4s | 4p | 4d |
|---|---|---|---|---|---|---|---|---|---|
| $\lambda_{nl}$ (SS) | 1.43544 | 1.13164 | 1.07454 | 1.08642 | 1.06983 | 1.07449 | 1.03921 | 1.08363 | 0.97834 |
| $\lambda_{nl}$ (AS) | 1.43540 | 1.13170 | 1.07456 | 1.08640 | 1.06982 | 1.07451 | 1.03930 | 1.08362 | 1.46072 |

In SS (Eissner et al., 1974) [9] and AS (Badnell, 2011) [12] atomic structure codes, the target wavefunctions are constructed using radial wavefunctions calculated in a scaled Thomas–Fermi–Dirac–Amaldi statistical model potential with a set of scaling parameters. The interactions of configurations give rise to 1014 fine structure levels.

## 3. Results and Discussion

We calculated powers of the radius $< r^n >, (n = -1, 1, 2$ and $3)$ for the first nine orbitals of Cr I using the three atomic structure codes CW, SS and AS and hydrogenic approximation (HA) method. We also compared the obtained values with those of Saito 2009 [13], which are calculated using the Hartree–Fock–Roothaan method. Tables 2–5 show the results of $< \frac{1}{r} >, < r >, < r^2 >$ and $< r^3 >$ for the different orbitals.

The values obtained by the three atomic structure codes are in good agreement in general, but hydrogen approximation (HA) gives good values only for the inner orbits. When comparing our calculations of $< \frac{1}{r} >$ from AS, SS and CW with values done by Saito 2009 [13], we see that values in all orbitals are in accordance (the error is only 2.25%) except for the $3d$ orbital, where the value of AS has scatter from others. Hydrogenic approximation (HA) gives good value for $1s$ orbital (error 2%), and, by moving to other far orbitals, the error will increase by getting away from the nucleus. The values of $\langle r \rangle$ have good agreement with the value of Saito with average error 2.3%, expect for the

3*d* orbital value of AS, which is three times the values in other calculations. While comparing $<r^2>$ values, it is observed that for the 3*d* orbit, the SS result is scattered from other values, and all other data match well (95%) with values of Saito 2009 [13]. In our particular case of the Cr I study and with the five first configurations used, the HFR method using the CW atomic structure code gives slightly better results compared with the two other AS and SS atomic structure codes. New values are also obtained for 4*p* and 4*d* of $<\frac{1}{r}>, <r>$ and $<r^2>$, which are not found by Saito 2009 [13]. For all the orbitals considered here, new values of $<r^3>$ are found using AS, SS and CW atomic structure codes.

Although the study of energy levels is in progress and as an example, we see that for the $3d^5(^6S)4p\ z^5P_1$ energy level (equal 26,902 cm$^{-1}$ in the NIST database [1]), the CW code gives 0.788% the NIST value, AS and SS codes give respectively 9.47% and 9.43% the NIST value. Thus, the HFR method gives better energy levels than the TFDA potential method and we recommend the use of the CW values as the most accurate ones for any physical or astrophysical use.

**Table 2.** Comparing the values of $\left\langle \frac{1}{r} \right\rangle$ calculated by the hydrogenic approximation (HA) and by the AS, SS, and CW atomic structure codes with values calculated by Saito 2009 [13] in a.u.

| *nl* | HA | AS | SS | CW | Saito |
|------|------|----------|----------|--------|-----------|
| 1*s* | 24 | 23.25061 | 23.54484 | 23.88 | 23.527266 |
| 2*s* | 6 | 4.87937 | 4.98681 | 5.086 | 5.013676 |
| 2*p* | 6 | 4.82134 | 4.92879 | 4.955 | 4.931330 |
| 3*s* | 2.67 | 1.45118 | 1.52920 | 1.55 | 1.529583 |
| 3*p* | 2.67 | 1.30494 | 1.40941 | 1.405 | 1.394535 |
| 3*d* | 2.67 | 0.56868 | 1.05602 | 0.985 | 1.03724 |
| 4*s* | 1.5 | 0.35246 | 0.37141 | 0.3341 | 0.346490 |
| 4*p* | 1.5 | 0.22883 | 0.28328 | 0.2705 | |
| 4*d* | 1.5 | 0.16087 | 0.09185 | 0.1083 | |

**Table 3.** Comparing the values of $\langle r \rangle$ calculated by the Hydrogenic approximation (HA) and by the AS, SS, and CW atomic structure codes with values calculated by Saito 2009 [13] in a.u.

| *nl* | HA | AS | SS | CW | Saito |
|------|--------|----------|----------|---------|----------|
| 1*s* | 0.0625 | 0.06498 | 0.06399 | 0.06351 | 0.064146 |
| 2*s* | 0.25 | 0.30217 | 0.29381 | 0.2911 | 0.294014 |
| 2*p* | 0.2083 | 0.26716 | 0.26052 | 0.2591 | 0.260079 |
| 3*s* | 0.5625 | 0.96420 | 0.90833 | 0.903 | 0.912446 |
| 3*p* | 0.5208 | 1.04902 | 0.97021 | 0.9714 | 0.977756 |
| 3*d* | 0.4375 | 3.25369 | 1.28118 | 1.412 | 1.367998 |
| 4*s* | 1 | 3.70946 | 3.44945 | 3.794 | 3.675128 |
| 4*p* | 0.9583 | 5.56900 | 4.65144 | 4.637 | |
| 4*d* | 0.8333 | 12.22154 | 16.77313 | 11.33 | |

**Table 4.** Comparing the values of $\langle r^2 \rangle$ calculated by the Hydrogenic approximation (HA) and by the AS, SS, and CW atomic structure codes with values calculated by Saito 2009 [13] in a.u.

| *nl* | HA | AS | SS | CW | Saito |
|------|---------|----------|-----------|----------|-----------|
| 1*s* | 0.0052 | 0.0055 | 0.0057 | 0.005425 | 0.005520 |
| 2*s* | 0.05208 | 0.1017 | 0.1080 | 0.1001 | 0.101988 |
| 2*p* | 0.01823 | 0.0835 | 0.0880 | 0.08247 | 0.083058 |
| 3*s* | 0.3594 | 0.9497 | 1.0759 | 0.9407 | 0.960479 |
| 3*p* | 0.3125 | 1.1085 | 1.3043 | 1.113 | 1.128512 |
| 3*d* | 0.2656 | 2.1699 | 17.1805 | 2.704 | 2.505944 |
| 4*s* | 1.111 | 13.8975 | 16.1200 | 16.77 | 15.722039 |
| 4*p* | 1.041 | 25.8320 | 36.4743 | 25.13 | |
| 4*d* | 0.875 | 320.3925 | 178.70952 | 145.4 | |

**Table 5.** Values of $\langle r^3 \rangle$ calculated by the Hydrogenic approximation (HA) and by the AS, SS, and CW atomic structure codes in a.u.

| $nl$ | HA | AS | SS | CW |
|------|-----|-----|-----|-----|
| 1$s$ | 0.0005425 | 0.0006 | 0.000589 | 0.0005836 |
| 2$s$ | 0.02387 | 0.0442 | 0.0402 | 0.03933 |
| 2$p$ | 0.01519 | 0.0347 | 0.0319 | 0.03128 |
| 3$s$ | 0.2490 | 1.3543 | 1.11 | 1.101 |
| 3$p$ | 0.2563 | 1.8721 | 1.45 | 1.466 |
| 3$d$ | 0.123 | 124.6413 | 4.74 | 6.800 |
| 4$s$ | 1.3541 | 80.5199 | 64.4 | 85.04 |
| 4$p$ | 1.2153 | 274.9271 | 167 | 273.3 |
| 4$d$ | 0.9479 | 2892.9064 | 6650 | 2080 |

## 4. Conclusions

In this work, expectation values of the radius are calculated for neutral chromium (Cr I) for the first five configurations. The calculated data are useful for interpretation of laboratory and astrophysical spectra. The expectation values of the neutral chromium (Cr I) radius are carried out with four different methods: SUPERSTRUCTURE code (SS), AUTOSTRUCTURE code (AS), Cowan code (CW) and hydrogen approximation (HA). The values of the three atomic structure codes have good agreement in general with the values of Saito 2009 [13]. For the neutral chromium and after comparing the different method results, we recommend the use of the CW values as the most accurate ones for any atomic structure data needs. Hydrogenic approximation data are approved only for the inner orbits. New values of $< \frac{1}{r} >, < r >$ and $< r^2 >$ for the configuration of 4$p$ and 4$d$ and the values of $< r^3 >$ for all of the first nine orbitals are obtained.

**Author Contributions:** N.A.R.Y. and N.B.N. performed the calculations and prepared the manuscript; all authors participated in the discussion.

**Funding:** This research received no external funding.

**Acknowledgments:** This research project was supported by a grant from the "Research Centre of the Female Scientific and Medical Colleges", Deanship of Scientific Research, King Saud University.

**Conflicts of Interest:** The authors declare no conflict of interest.

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
