# Peer review of "Expectation Values of the Neutral Chromium Radius"

_atoms, doi:10.3390/atoms6030051_

Round 1
Reviewer 1 Report
The authors report-on a rather narrow study of radial expectation values for neutral Cr, utilizing several codes. Certainly, Cr occurs in some fusion experiments and motional Stark effects maybe of interest. The calculations are rather straightforward using publicly available codes. I have a few comments: 1/ You should reference the original literature when talking about the additional operators in AS compared to SS. I assume you mean two-body non-fine-structure (Badnell, JPB3 v0, p.1, 1997)? 2/ You need to detail how you obtained different results from SS and AS. Using the same configurations, Thomas-Fermi potentials, and operators, SS and AS give identical results since SS is a subset of AS. You cannot optimize SS in intermediate coupling, contrary to what is stated in the MS. 3/ Given 2/, it would be useful to give the scaling parameters used for SS and AS so that others can reproduce the results, and obtain further data such as radiative rates. 4/ Given the scatter of results which sample the outer part of the wavefunction, e.g. , which do you recommend as the most accurate, and why? 5/ Given 4/, a comparison of energy levels with observed should tell you which are more reliable wavefunctions, although not necessarily at all r.Author Response
1-Corrected by using the original reference (Badnell 1997) (see page 2, line 47).
2- Corrected: The optimization is for the sum of all the energies of all the target terms. The text was:
“The radial functions are calculated in scaled Thomas-Fermi statistical model potential which depends on parameters called λnl, those n and l dependent scaling parameters λnl were determined variationally.
In SS ( Eissner (1974))[4] and AS code (Badnell 1997)[8]. Which constructs target wavefunctions using radial wavefunctions calculated in a scaled Thomas- Fermi-Dirac-Amaldi statistical model potential with a set of scaling parameters.
The interactions of configurations gives rise to 1014 fine structure levels. The radial scaling parameters λnl are determined by minimizing the sum of the energies of all the target terms, computed in intermediate coupling. Table 1 give the scaling parameters for the potentials in which the orbital functions are calculated.”
Becomes (see page 2, lines 48-53):
“The radial functions are calculated in scaled Thomas-Fermi statistical model potential which
depends on parameters called λnl . Those n and l dependent scaling parameters λnl were determined variationally by minimizing the sum of the energies of all the target terms.
The scaling parameters λnl obtained in the SS and AS atomic structure codes are given in Table 1. The difference between the values is caused by a small change in the optimization procedure in AS comparing to the SS procedure.”
(This small change in procedure started from version 20.20 of AS)
3- The scaling parameters used in SS and AS are reported in Table 1 (see page 2, line 53-54).
4 &5- We recommend the CW results which are in our case better than the SS and AS values. We added, in page 2, section “3. Results and discussion”, the modifications in bold (page 4, line 71-83):
“While comparing < r2 > values, it is observed that for the 3d orbit, the SS result is scattered from other values, and all other data are in good match (95%) with values of Saito 2009 [12]. In our particular case of the Cr I study and with the five first configurations used, HFR method using the CW atomic structure code gives a slightly better results comparing to the two other AS and SS atomic structure codes. New values are also obtained for 4p and 4d of < 1/r >,< r > and < r2 > which are not found by Saito 2009 [12]. For all the orbitals considered here, new values of < r3 > are found using AS, SS and CW atomic structure codes.
Even the study of energy levels is in progress and as an example, we see that for the 3d5(6S)4p z5P1 energy level (equal 26902 cm-1 in NIST database [1]), the CW code gives 0.788 % the NIST value, AS and SS codes give respectively 9.47 and 9.43 % the NIST value. The same remark for the 3d44s(5D)4p z3P1 energy level (equal 33897 cm-1 in NIST database [1]). So, the HFR method gives better energy levels than the TFDA potential method and we recommend the use of the CW values as the most accurate one for any physical or astrophysical use.”.
We also added the following text in the conclusion before the last sentence:
For the neutral chromium and after comparing the different method results, we recommend the use of the CW values as the most accurate one for any atomic structure data need.

Reviewer 2 Report
Reviewer comments
Manuscript ID: atoms-337491
Type of manuscript: Article
Title: Expectation Values of the Neutral Chromium Radius
Authors: Nafeesah Abdul Rahim Yaqub, Rabia Qindeel, Norah Alonizan, Nabil Ben
Nessib *
In this paper authors presents expectation values of the neutral chromium radius. These atomic data are calculated with different atomic codes. Obtained results are important for spectral line broadening and as such needed for the laboratory research and for the astrophysical investigation.
The manuscript represents original contribution and it is clearly written. I think results will be interesting for potential readers. I would like to recommend publications of this paper in Atoms but with few requests.
Some requests:
The authors did not type the text in the MDPI template and manuscript have no line numbers, so it's difficult to tag and indicate where corrections are needed.
-Author affiliations:
Correspondence: [email protected]; Tel.: +966...=> [email protected]
or give the whole phone number.
Sec. 2 (second paragraph): Eissner et al. [8] => Eissner et al. 1974 [8]
or Eissner and coworkers [8]
Sec. 2 (at the end of the sec.2): (Alonizan et al., [10]). => (Alonizan et al. 2016, [10]).
- Authors should extend Introduction with sentence or two about why the obtained results for chromium are important. This would indicate the reason for accepting this work and it will be good for potential readers. Currently only one sentence describes this issue “Chromium in its various ionization stages is important for analyzing atmospheres of some stars and many works are done for the chromium in astrophysical plasmas (see [3]).”
-Also, how the differences in values obtained in this paper by different codes (values in tabs.) affect the Stark broadening calculations. What is their assessment. It does not have to be calculated but to be estimated. It would be interesting for readers.
All references should be written in the same way, according to the journal style.
e.g.
[6] Hamdi R., Ben Nessib N., Sahal-Brechot S. and Dimitrijevic M. S., Atoms, 2017, 5, 26
=>
[6] Hamdi, R.; Ben Nessib, N.; Sahal Bréchot, S. & Dimitrijević, M. S. Stark Widths of Ar II Spectral Lines in the Atmospheres of Subdwarf B Stars. Atoms, 2017, 5(3), 26.
….
and, also for the books…
- In some refs. authors use full journal name and on some abbreviation. They should be written in the same way.
Respectfully,
Author Response
1- Author affiliation is corrected.
2- The text style, citations and the references are done using the Latex template form, the Tables are in their places and not at the end of the document.
3- Introduction is extended by the 2 sentences in page 1, lines 18-20 (We added the Sobeck 2007 reference):
For laboratories and astrophysical plasmas, the study of chromium is important, it occurs in some fusion experiments. Many laboratory techniques are used to have Cr I atomic structure data
(see Sobeck et al. 2007 [3]).
4- How the difference in values affect the Stark broadening calculations is explained briefly:
We added at the end of the Introduction the text (page 1, lines 26-29):
The contribution of the dipolar interaction in the Stark broadening is directly proportional to square orbital radius and a variation of it can affect directly the result of widths and shifts of the spectral lines broadened by Stark effect. Orbital radius and their square are also in the contribution of the quadrupolar interaction.
This is a simplification of the need of r and r2 in the Stark broadening calculations (which we usually do in cooperation with Meudon and Belgrade Observatories) more details will be out of the objectives of the work.

Round 2
Reviewer 1 Report
Suitable changes have been made.